

# A feature boosted deep learning method for automatic facial expression recognition

Tanusree Podder[1], Diptendu Bhattacharya[1], Priyanka Majumder[2] and Valentina Emilia Balas[3]

[1] Department of Computer Science and Engineering, National Institute of Technology Agartala, Agartala, Tripura, India
[2] Department of Basic Science and Humanities, Techno College of Engineering Agartala, Agartala, Tripura, India
[3] Department of Automation and Applied Informatics, Aurel Vlaicu University of Arad, Arad, Romania

## ABSTRACT

Automatic facial expression recognition (FER) plays a crucial role in human-computer based applications such as psychiatric treatment, classroom assessment, surveillance systems, and many others. However, automatic FER is challenging in real-time environment. The traditional methods used handcrafted methods for FER but mostly failed to produce superior results in the wild environment. In this regard, a deep learning-based FER approach with minimal parameters is proposed, which gives better results for lab-controlled and wild datasets. The method uses features boosting module with skip connections which help to focus on expression-specific features. The proposed approach is applied to FER-2013 (wild dataset), JAFFE (lab-controlled), and CK+ (lab-controlled) datasets which achieve accuracy of 70.21%, 96.16%, and 96.52%. The observed experimental results demonstrate that the proposed method outperforms the other related research concerning accuracy and time.

## INTRODUCTION

Automatic identification of a human being's mental state is a critical task for a machine because of the structure of the human face. However, most automated emotion detection processes take 'face' as their input despite the complex structure of a human face. According to the Mehrabian rule (*Mehrabian & Ferris, 1967*), human emotion depends 7% on spoken words, 38% on vocal tone, and 55% on facial expression. This finding emphasizes that facial expression is essential to the nonverbal conversation and the primary means to convey emotions. Hence, most researchers select static images for extracting the facial information, but methods of 3D, multi-modal (*Tzirakis et al., 2021*), and thermal (*Nayak et al., 2021*) are also available in the literature.

The conventional approaches of FER first detect faces and extract handcrafted features to feed empirical machine learning classifiers like SVM, Adaboost, *etc.* In recent years, researchers have mainly focused on CNN-based (*Gan et al., 2022*; *Tong, Sun & Fu, 2022*) methods for detection (*Appati, Adu-Manu & Owusu, 2022*; *Owusu, Appati & Okae, 2022*)

Corresponding authors
Tanusree Podder,
tanusreepodder29@gmail.com
Valentina Emilia Balas,
balas@drbalas.ro

and classification, where the learning takes place by itself, without any handcrafted feature extraction techniques. However, CNN requires a large amount of training data to prevent overfitting, so the ideal dataset size for CNN should be in the millions. Further, the memory requirement and computation power needed for CNN-based methods are higher than conventional FER methods. Besides, some real-time applications (*Miao et al., 2019*) only detect emotions from the frontal face as they used a lab-controlled dataset (*e.g.*, CK+) for training. Therefore, the dataset's versatility also plays a vital role in improving it while considering CNN as the feature extractor.

Motivated by the above discussion, a deep learning model is proposed for FER in the lab-controlled and wild environment. The model is designed with minimal parameters to reduce the detection time. Further, it applies pre-trained weight from FER-2013 to CK+ and JAFFE, as their dataset size is less than 1,000. For real-time detection, the proposed method uses the trained model of FER-2013 since it contains approximately 30K images from wild environments. In brief, the principal contributions of this study are as follows:

a) We propose a novel CNN structure for FER by introducing a feature-boosting block that focuses on expression-specific features with the help of dense connections and a translation block that reduces redundancy and dimensionality.

b) Data augmentation is employed in FER-2013 to add versatility to the dataset. In addition, transfer learning is applied while training small datasets, which saves training time and improves performance.

c) Further, an application is developed to detect real-time facial expressions with a lower runtime cost than the literature.

Following is a summary of the remainder of the article: "Literature Survey" briefly analyzes the traditional and current facial expression recognition work. "Proposed Methodology" explains the entire methodology adopted for the training process. A detailed analysis of experiments and findings is presented in "Datasets". A summary of the advantages and directions for future work is provided in "Conclusion".

## LITERATURE SURVEY

FER-based methods can generally be classified into two categories based on the feature extraction process, namely, handcrafted and deep learning methods. The handcrafted feature-based methods can be further subdivided into appearance (*Happy & Routray, 2014*; *Lekdioui et al., 2017*), geometric (*Vishnu Priya, 2019*), and hybrid methods, which are commonly used in traditional FER approaches. In appearance-based methods (*Yan et al., 2020*), features are derived from intensity, gradient, and texture variations of identified facial regions. Geometric feature extraction, however, relies on structural information about the face. For instance, distance estimations from the fiducial points of the face can be used to identify facial expressions. For FER, a new feature descriptor Cross-Centroid Ripple Pattern (*Verma et al., 2022*) is proposed by incorporating a cross-centroid relationship between two ripples. By subdividing the local neighborhood region into sub-

regions, ripples are generated. Additionally, gradient information between ripples provides the ability to capture prominent edge features in active sub-regions.

On the other hand, the hybrid method combines appearance and geometric methods to extract features. For the final classification, the handcrafted features must be fed into traditional classifiers such as neural networks (NN), SVM, Adaboost, *etc.* The handcrafted methods usually perform well in lab-controlled datasets because they are defined mainly by prior knowledge. A hybrid method for feature extraction is utilized to detect emotion from static images and video frames (*Khan et al., 2019*). The method extracted 24 fiducial points to calculate sixteen mutual distances to use as a geometric feature. Moreover, an optical flow-based method was used for video emotion recognition and to track each of the 24 points over eight frames. However, emotions like anger and disgust require appearance-based features and mutual distances to correctly classify facial expressions. For that, the features from the region between the eyebrows and wrinkles of the nose were extracted. Further, the dimension was reduced using PCA before feeding into the probabilistic neural network (PNN). The resultant information from PNN passes through a bagging layer for final classification.

On the other hand, deep learning methods have been successfully utilized in lab-controlled and unconstrained environments to achieve remarkable performance. Recent studies have shown that it outperforms handcrafted methods (*Liu et al., 2014*) in many areas, such as image classification and computer vision. It is the self-learning capability of CNN to extract features automatically and the development of deep learning models such as AlexNet, ResNet, MobileNet, VGG16 (*Simonyan & Zisserman, 2014*), VGG19, Inception V3 (*Szegedy et al., 2016*), and DenseNet (*Huang et al., 2017*) that attracted the researcher to work in this domain. The researchers modified these deep learning models to apply in FER. For instance, AlexNet is modified in *Fei et al. (2020)* to analyze patients' mental health conditions using FER. Similarly, the VGG-16 model is utilized in *Ahmed et al. (2018)* to detect facial expressions in the wild environment with an incremental active learning framework.

However, training a deep learning model is complex and time-taking and requires large datasets and high processing power to produce optimal results. Furthermore, most existing FER datasets generally contain hundreds or thousands of samples. Therefore, overfitting becomes a significant challenge that requires to be addressed. As a solution, the generative model has been applied to create new images for FER using the existing ones (*Han & Huang, 2021*). Another solution is to use transfer learning, where the pre-trained weight of existing CNN models with a large dataset is applied to a small dataset by changing the last few layers. For example, *Miao et al. (2019)* used two rounds of transfer learning ImageNet to FER-2013 and FER-2013 to CK+ using MobileNet as CNN architecture. Similarly, the work of *Hung, Lin & Lai, (2019)* utilizes two rounds of transfer learning from JAFFE and KDEF to FER-2013 and vice versa to obtain good accuracy using their own CNN model. Instead of using the inbuilt deep learning models, researchers proposed different CNN architectures (*Xie, Hu & Wu, 2019*; *Shao & Qian, 2019*) to solve the FER problem. For instance, *Jain, Shamsolmoali & Sehdev (2019)* designed a Deep CNN with convolution layers and residual blocks to detect six types of facial emotion. They applied the CK+ and

JAFFE datasets to carry out the experiments. Further, in *Lopes et al. (2017)*, specific image pre-processing steps (cropping, spatial normalization, and synthetic samples) are combined with CNN for FER. Facial expression detection becomes more challenging due to inter-class similarities and intra-class variations. *Fard & Mahoor (2022)* proposed an adaptive correlation loss function which generates an embedded feature vector using three types of discriminators. The Xception network is used as the backbone of the CNN model.

A lightweight CNN model for mental health monitoring in a cloud-based environment is proposed in *Podder, Bhattacharya & Majumdar (2022a)*. However, in *Saurav et al. (2022)*, a dual-integrated lightweight model is deployed for real-time facial expression recognition in wild environments. Recently, *Gera & Balasubramanian (2021)* proposed an attention-based end-to-end method to combine local and global facial features without using landmark information. Further, it uses complementary context information to boost the detection rate. Contrary to all, some novel data augmentation techniques are proposed in *Umer et al. (2022)* to generate versatile images to train the CNN model. Finally, the CNN model has been fine-tuned by trading off data augmentation and deep learning features. The effectiveness of data augmentation in occluded faces is proposed in *Asiedu et al. (2021)*.

After reviewing the above literature, the following reasons encouraged us to design a new CNN architecture: First of all, most deep learning models prefer an input size of $224 \times 224$ (*Simonyan & Zisserman, 2014*; *Huang et al., 2017*; *Ahmed et al., 2018*), $299 \times 299$ (*Szegedy et al., 2016*) or $256 \times 256$ (*Fei et al., 2020*) which is larger than the image size of FER-2013 ($48 \times 48$). Therefore, to train the model with FER-2013, the images need to be resized, which adds redundant information, leading to repetitive feature learning. Secondly, pre-trained weights on ImageNet are available for deep learning models in Keras Applications (KA), but it requires a colour image as input. In the case of a greyscale image, KA repeats the channel thrice to convert it into a colour image, resulting in unnecessary computation with redundant data since FER-2013 contains greyscale images. Thirdly, the memory requirements of available deep learning models motivated us to design CNN architecture with fewer parameters to save memory and running time.

## PROPOSED METHODOLOGY

A detailed overview of the proposed methodology has been depicted in Fig. 1. The proposed method first detects the face and performs the greyscale conversion and resizing as preprocessing steps. After that, training data is augmented to add versatility which is then trained with the proposed deep CNN model. The best-trained model is applied to classify the expressions into seven universal categories. The description of the various component of the proposed methodology are described below:

### Preprocessing

Pre-processing is a crucial step to follow before the model training. Initially, as a preprocessing step, the face is detected using the Haar cascade classifier proposed by *Viola & Jones (2004)*. Following this, greyscale conversion and resizing the image to $48 \times 48$

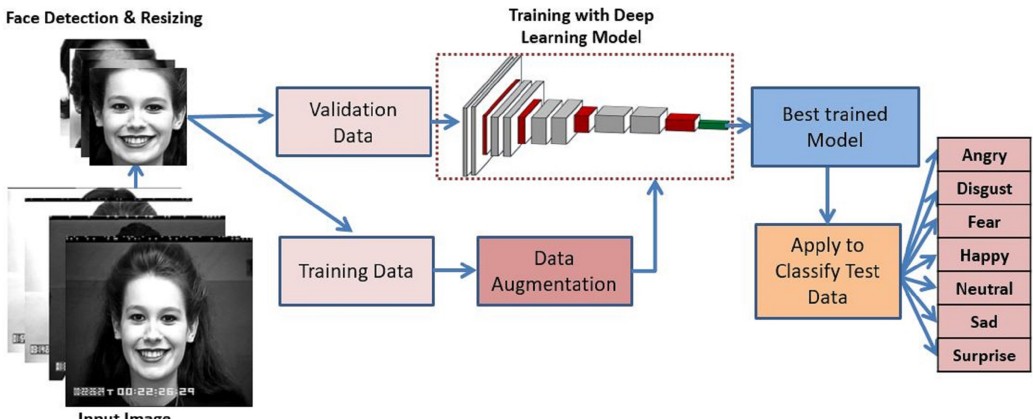

**Figure 1 Proposed system model for FER.** Image source credit: S74, the CK and CK+ dataset, *Lucey et al. (2010)* © Jeffrey Cohn.

pixels is carried out to remain consistent with the size of the FER-2013 dataset. In addition, on-the-fly data augmentation is conducted before model training to artificially increase the dataset variation. In the augmentation process, the width and height shift with 0.3, zoom and shear range with 0.3, horizontal flip, rotations of 30°, rescale to [0,1] and fill mode as nearest is applied.

## The CNN architecture

After preprocessing, the facial image is fed into the CNN. The proposed CNN architecture is shown in Fig. 2. The CNN architecture starts with a convolution (Conv) layer. It helps to extract hierarchical features by convolving image patches with filters of different kernel sizes. Equation (1) below shows the mathematical calculation of CL:

$$X_j^l = F\left(\sum_{i=1}^{M_{l-1}} X_i^{l-1} * W_{ij} + W_b\right) \tag{1}$$

Here, $*$ denotes convolution operation; $X_i^{l-1}$ represents feature maps of $l - 1$ layer convolving with filter $W_{ij}$ and offset term is represented as $W_b$. Initially, the 1st Conv layer extracts 32 feature maps from a $48 \times 48$ input image with 32 kernels of size $3 \times 3$. After that, the 2nd Conv layer extracts 32 feature maps from the output of the 1st Conv layer with the same kernel size, as shown in Fig. 2. A ReLU activation function follows every Conv layer due to its interaction effect and nonlinear properties. The function returns the value it receives if the input is positive; otherwise, it returns 0. It can be calculated using Eq. (2), where z is the input neuron.

$$f(z) = \max(0, z) \tag{2}$$

The feature maps obtained through the Conv layer are preceded by the max-pooling layer that removes redundant information and sums up the presence of features in patches of a feature map by down-sampling it. The calculation of the pooling layer is given in Eq. (3) below:

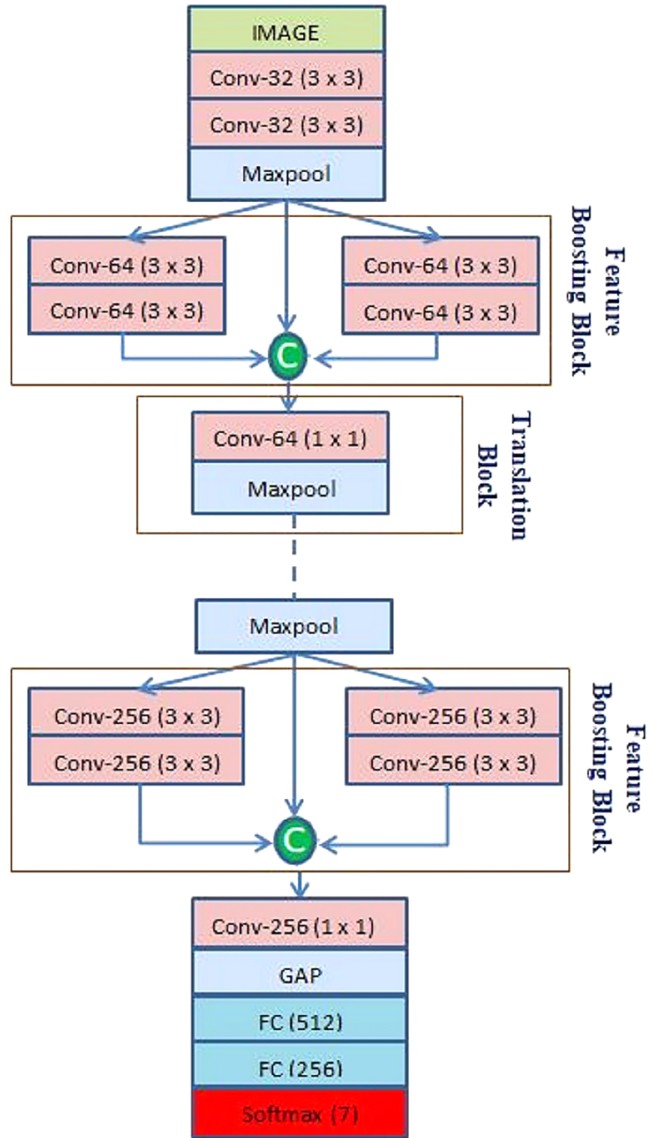

**Figure 2 Architecture of the proposed CNN model.**

$$X_j^l = F\left(MP(X_j^{l-1}) + W_b\right) \qquad (3)$$

where, $j^{\text{th}}$ feature map of pooling layer $l-1$ denoted as $X_j^{l-1}$ and offset term of pooling layer represented as $W_b$.

The feature maps from the max pooling layer is passed through two branches each containing two consecutive Conv layers. In addition, for better gradient propagation a skip connection is used to propagate the features of the max pooling layer. Finally, all these feature maps are concatenated to prevent feature loss while boosting the feature collection. Let $X = \{A_{i,i=1\ldots2^a}\}$, $X_1 = \{B_{j,j=1\ldots2^{a+1}}\}$ and $X_2 = \{C_{k,k=1\ldots2^{a+1}}\}$ be the features extracted through the max pooling layer and two branches of the Conv layer. The obtained

feature map $F_C$ is passed to the translation block. Here, $2^a$ is the no of feature maps obtained after max-pooling layer.

$$F_C = Concate[X, X_1, X_2] = |A_1, ...., A_{2^a}, B_1, ...., B_{2^{a+1}}, C_1, ...., C_{2^{a+1}}| \qquad (4)$$

In the translation block first, $1 \times 1$ Conv layers are applied to extract information which is followed by a max pooling layer for dimensionality reduction. The proposed model adds three such feature boosting and translation blocks while the feature map sizes are increased from 64 to 128, and 256, as shown in Table 1. At the end of the final translation block instead of using a max pooling layer, a global average pooling layer is used. Lastly, the extracted features are passed to two consecutive fully connected (FC) layers and classified with a softmax activation function. It converts the input into probabilities by calculating the exponents and normalizing each input link with the sum of all as presented in Eq. (5)

$$S(x_i) = \frac{e^{x_i}}{\sum\limits_i e^{x_i}} \qquad (5)$$

Here, $x_i$ represent individual elements of each class.

Batch normalization is also utilized to normalize the output of the previous layers for each mini-batch. It has the impact of stabilizing the learning process by drastically reducing the number of training epochs required to train the network. The network uses the dropout layer, which randomly sets a percentage of activations links to zero, thereby helping to identify dissimilar data. The model added a dropout layer with a probability of 0.2 after the first block and each translation block. Further added two dropout layers after each Fully connected layer. Table 1 gives a detailed overview of the input and output given to a particular layer, along with the filter size, shape and stride used in designing the architecture.

Figure 3 shows the Grad-CAM visualization results generated from the input image for the $1^{st}$, $2^{nd}$, $6^{th}$, $8^{th}$, $10^{th}$, $18^{th}$ and $26^{th}$ Conv layers. Observations from Fig. 3 show that different Conv layers of the model extract discriminated information for facial expression classification by focusing on different areas of the face. The $10^{th}$, $18^{th}$ and $26^{th}$ Conv layers convolve features boosted from the lower Conv layers, *e.g.*, in $26^{th}$ Conv layer, the integrated features from layer $20^{th}$, $22^{nd}$ and $24^{th}$ is applied as input. The initial Conv layers focus on overall facial features but the Conv layer applied after integrating features like the $10^{th}$, $18^{th}$ and $26^{th}$ focus on expression-specific action units (AU). *E.g.*, AU6 (check raiser) and AU12 (lip corner puller) are focused on the expression smile in the higher Conv layer ($18^{th}$, $26^{th}$), as shown in Fig. 3.

## DATASETS

The two extensively used standard datasets, CK+ (*Lucey et al., 2010*) and JAFFE (*Lyons et al., 1998*) have been chosen for evaluation. In addition, another dataset, FER-2013 (*Goodfellow et al., 2013*), is selected in this work due to its largest size, versatility, and free availability. Figure 4 shows examples of basic facial expressions from three datasets. The

**Table 1 Parameter details of the proposed model.**

| Layer no. | Layer name | Filter shape | No. of filter | Stride | Input1 | Input2 | Input3 | Output |
|---|---|---|---|---|---|---|---|---|
| 1 | Conv-32 | 3 × 3 | 32 | 1 | 48 × 48 × 1 | – | – | 48 × 48 × 32 |
| 2 | Conv-32 | 3 × 3 | 32 | 1 | 48 × 48 × 32 | – | – | 48 × 48 × 32 |
| 3 | Maxpool | 2 × 2 | – | 2 | 48 × 48 × 32 | – | – | 24 × 24 × 32 |
| 4 | Dropout (0.2) | – | – | – | – | – | – | – |
| 5 | Conv-64 | 3 × 3 | 64 | 1 | 24 × 24 × 32 | – | – | 24 × 24 × 64 |
| 6 | Conv-64 | 3 × 3 | 64 | 1 | 24 × 24 × 64 | – | – | 24 × 24 × 64 |
| 7 | Conv-64 | 3 × 3 | 64 | 1 | 24 × 24 × 32 | – | – | 24 × 24 × 64 |
| 8 | Conv-64 | 3 × 3 | 64 | 1 | 24 × 24 × 64 | – | – | 24 × 24 × 64 |
| 9 | Concate (4, 6, 8) | – | – | – | 24 × 24 × 32 | 24 × 24 × 64 | 24 × 24 × 64 | 24 × 24 × 160 |
| 10 | Conv-64 | 1 × 1 | 64 | 1 | 24 × 24 × 160 | – | – | 24 × 24 × 64 |
| 11 | Maxpool | 2 × 2 | – | 2 | 24 × 24 × 64 | – | – | 12 × 12 × 64 |
| 12 | Dropout (0.2) | – | – | – | – | – | – | – |
| 13 | Conv-128 | 3 × 3 | 128 | 1 | 12 × 12 × 64 | – | – | 12 × 12 × 128 |
| 14 | Conv-128 | 3 × 3 | 128 | 1 | 12 × 12 × 128 | – | – | 12 × 12 × 128 |
| 15 | Conv-128 | 3 × 3 | 128 | 1 | 12 × 12 × 64 | – | – | 12 × 12 × 128 |
| 16 | Conv-128 | 3 × 3 | 128 | 1 | 12 × 12 × 128 | – | – | 12 × 12 × 128 |
| 17 | Concate (12, 14, 16) | – | – | – | 12 × 12 × 64 | 12 × 12 × 128 | 12 × 12 × 128 | 12 × 12 × 320 |
| 18 | Conv-128 | 1 × 1 | 128 | 1 | 12 × 12 × 320 | | | 12 × 12 × 128 |
| 19 | Maxpool | 2 × 2 | – | 2 | 12 × 12 × 128 | – | | 6 × 6 × 128 |
| 20 | Dropout (0.2) | – | – | – | – | – | – | – |
| 21 | Conv-256 | 3 × 3 | 256 | 1 | 6 × 6 × 128 | – | | 6 × 6 × 256 |
| 22 | Conv-256 | 3 × 3 | 256 | 1 | 6 × 6 × 256 | – | | 6 × 6 × 256 |
| 23 | Conv-256 | 3 × 3 | 256 | 1 | 6 × 6 × 128 | – | | 6 × 6 × 256 |
| 24 | Conv-256 | 3 × 3 | 256 | 1 | 6 × 6 × 256 | – | | 6 × 6 × 256 |
| 25 | Concate (20, 22, 24) | – | – | – | 6 × 6 × 128 | 6 × 6 × 256 | 6 × 6 × 256 | 6 × 6 × 640 |
| 26 | Conv-256 | 1 × 1 | 256 | 1 | 6 × 6 × 640 | | | 6 × 6 × 256 |
| 27 | Dropout (0.2) | – | – | – | – | – | – | – |
| 28 | GAP | – | – | – | 6 × 6 × 256 | – | | 256 |
| 29 | FC | – | – | – | 256 | | | 512 |
| 30 | Dropout (0.5) | – | – | – | – | – | – | – |
| 31 | FC | – | – | – | 512 | | | 256 |
| 32 | Dropout (0.5) | – | – | – | – | – | – | – |
| 33 | Softmax | – | – | – | 256 | | | 7 |

expressions in FER2013 are from an uncontrolled environment, while CK+ and JAFFE datasets are lab-controlled.

*CK+:* The CK+ dataset was made available to the research community in 2010. It is an extension of the CK dataset with an increased no of subjects and sequences. It consists of the facial expression of 213 adults displaying neutral to peak emotion. The dataset includes posed images of six universal emotions with contempt. The images of the dataset are FACS-coded.

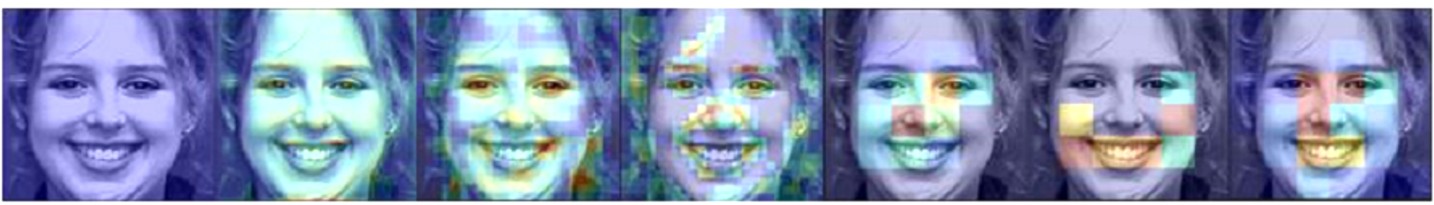

**Figure 3** **Grad-CAM visualization results of the proposed model.** Image source credit: S74, the CK and CK+ dataset, *Lucey et al. (2010)* © Jeffrey Cohn.

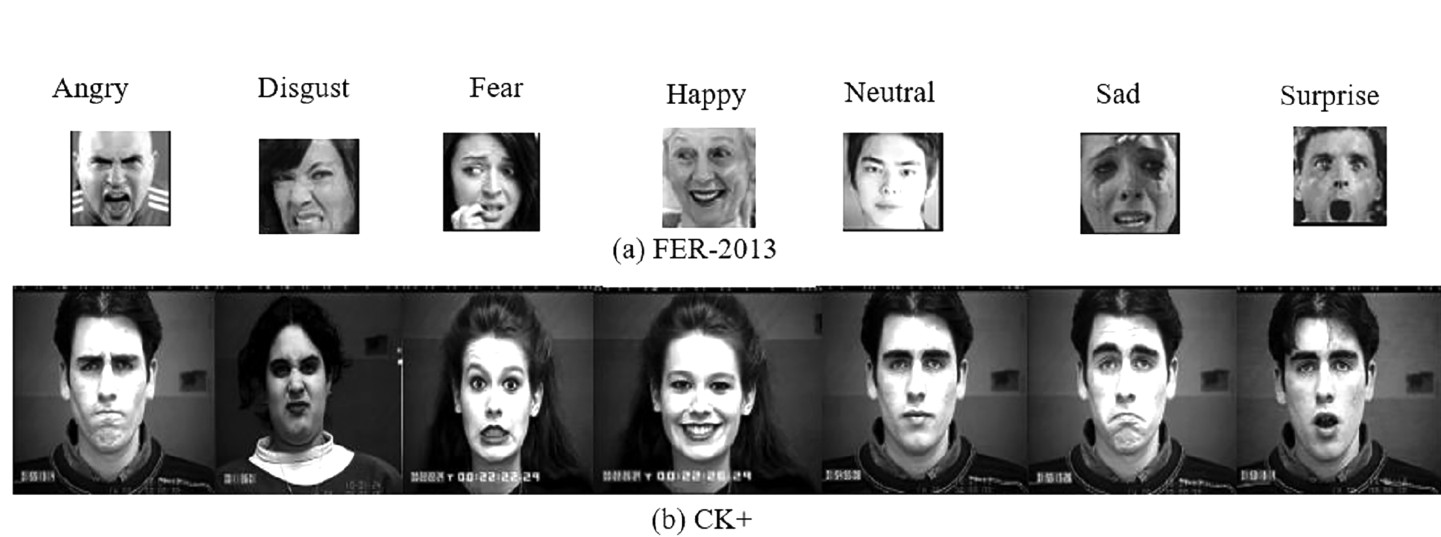

(a) FER-2013

(b) CK+

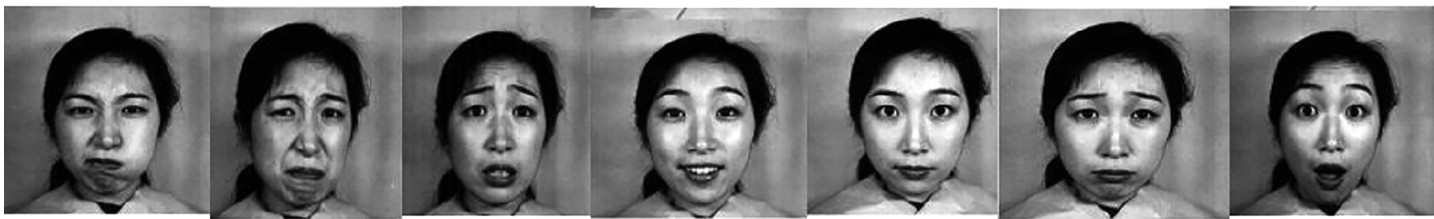

(c) JAFFE

**Figure 4** **Images of basic facial expression from FER-2013, CK+ and JAFFE. All the datasets contain seven basic emotional classes except CK+, which contains one extra class called contempt.** Image source credits: *Goodfellow et al. (2013)*; S74, S113, S124, the CK and CK+ dataset, *Lucey et al. (2010)*, © Jeffrey Cohn; the JAFFE dataset, *Lyons, Kamachi, Gyoba (2020)* and *Lyons (2021)*.

*JAFFE:* The dataset contains 213 grayscale images of six universal emotions with a neutral state. The size of the greyscale images is 256 × 256, and 10 female Japanese models posed them.

*FER-2013:* FER-2013 dataset contains 28,709, 3,589, and 3,589 images for training, validation, and testing. For training deep learning models, it's one of the best datasets available. Google's image search API was used to collect 35,887 images representing seven emotional levels. The images are 48 × 48 pixels each.

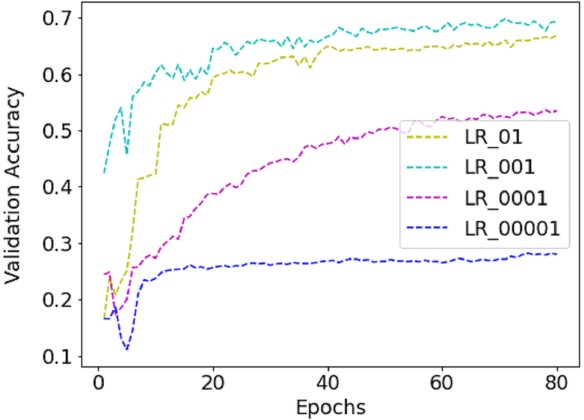

**Figure 5** **The performance of validation accuracy on FER-2013 dataset with different LR.**

## RESULTS AND DISCUSSIONS

While training the FER-2013 dataset, a batch size of 256 is selected as it contains nearly 30K images. Unlike FER-2013, CK+ and JAFFE are not distributed into training, validation and testing sets and contain 654 and 213 images only. So, five-fold and ten-fold cross-validation techniques are applied to avoid error due to random or self-selected train-test split. Due to this, the validation size became different for different folds, *e.g.*, for the JAFFE dataset, it varies from 21 to 22. So, for CK+ and JAFFE datasets, the batch size equals the validation set. The work utilizes the runtime environment of Google Colab to accelerate the training process as it provides Nvidia K80/T4 GUP support with upgraded RAM of size 26.75 GB. The inbuilt API service available in Keras 2.4.3 with TensorFlow 2.4.1 as a backend is employed for model creation and compilation. For model compilation, the Adam optimizer is applied with an initial learning rate (LR) of 0.001, which is dropped by 0.5 with every 20 epochs. The model is trained with different LR to find the best one, as shown in Fig. 5. Finally, 0.001 is selected because the validation accuracy is highest with it.

### Experimental results on FER-2013 dataset

Due to the diversity of the FER-2013 dataset, achieving reasonable accuracy with fewer preprocessing steps and light CNN architecture becomes a challenging task. Nevertheless, the augmentation process helps to increase accuracy by 1.98%. Table 2 illustrates how the proposed method outperforms different state-of-the-art CNN based models. Figures 6 and 7 shows the plots of training *vs* validation accuracy and loss up to 150 epochs and normalized confusion matrix. However, a section of the fear images is misclassified as sad and angry in the normalized confusion matrix due to small inter class differences.

### Experimental results on JAFFE dataset

The JAFFE dataset contains 213 grayscale images posed by 10 female Japanese models. As the number of images is less, the ten-fold cross-validation technique is applied to evaluate the accuracy. Further, the pre-trained weight on FER-2013 is used to accelerate the convergence rate. Figure 8A shows the normalized confusion matrix on validation data. It
**Table 2  Performance comparison with literature on the FER-2013 dataset.**

| Methodology | Accuracy (%) |
| --- | --- |
| AlexNet + FC6 + LDA (*Fei et al., 2020*) | 56.40 |
| MobileNet with Softmax + Center loss (*Miao et al., 2019*) | 68.31 |
| LiveEmoNet (*Podder, Bhattacharya & Majumdar, 2022b*) | 68.96 |
| Dense_FaceLiveNet (*Hung, Lin & Lai, 2019*) | 70.02 |
| Lightweight CNN (*Podder, Bhattacharya & Majumdar, 2022a*) | 68.93 |
| DenseNet201 (*Huang et al., 2017*) | 68.52 |
| VGG16 (*Simonyan & Zisserman, 2014*) | 65.80 |
| InceptionV3 (*Szegedy et al., 2016*) | 68.62 |
| Proposed method | 70.21 |

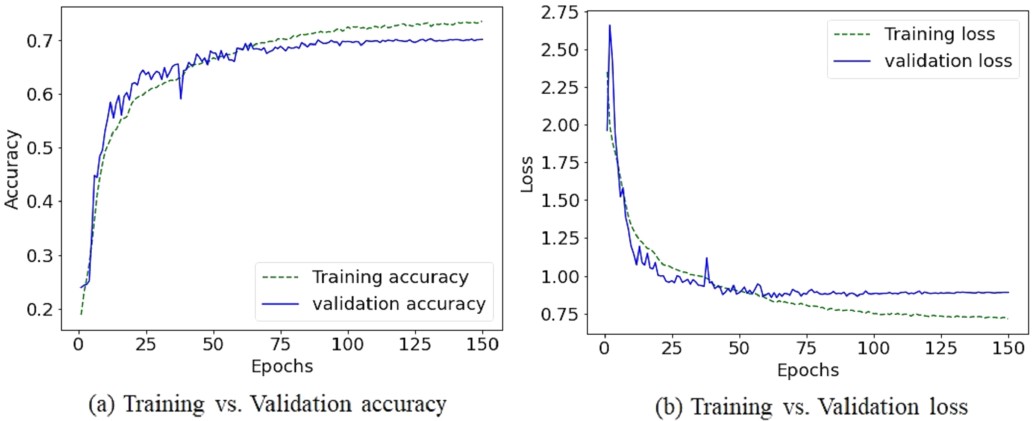

(a) Training vs. Validation accuracy    (b) Training vs. Validation loss

**Figure 6  The training *vs* validation accuracy and loss graph on FER-2013 dataset.**

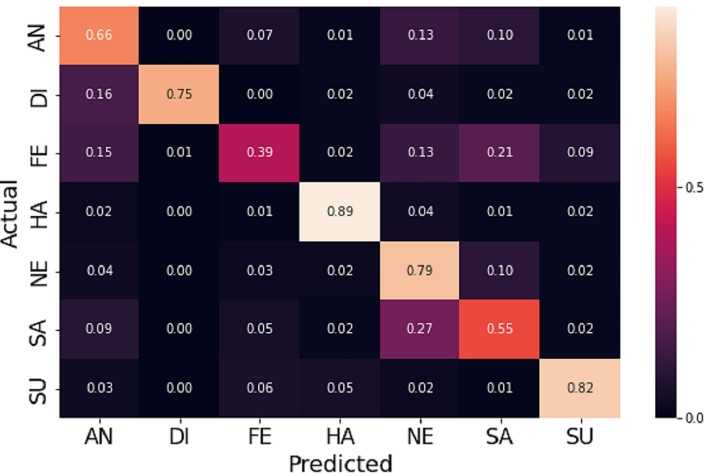

**Figure 7  The normalized confusion matrix on FER-2013 dataset.**

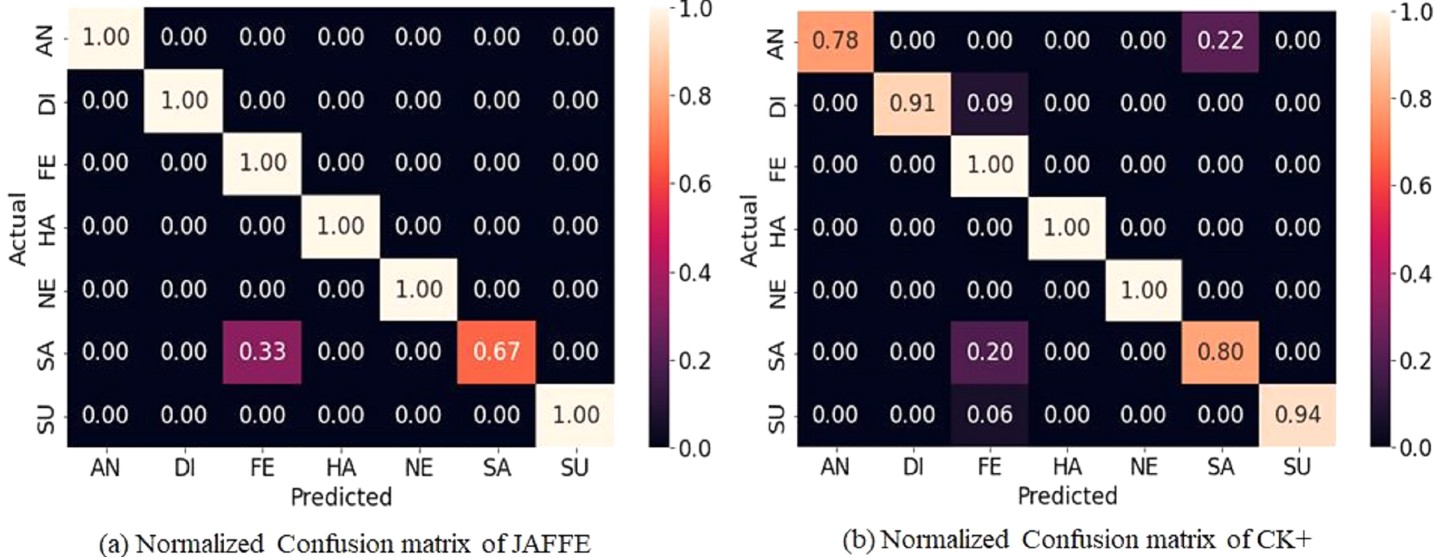

(a) Normalized Confusion matrix of JAFFE  (b) Normalized Confusion matrix of CK+

**Figure 8 Normalized confusion matrices for JAFFE and CK+ dataset.**

**Table 3 Performance comparison with literature on the JAFFE dataset.**

| Methods | Evaluation strategy | Accuracy (%) |
|---|---|---|
| Pre-processing + CNN (*Lopes et al., 2017*) | 10-fold | 37.36 |
| IFSL (SVM) (*Yan et al., 2020*), | 10-fold | 88.2 |
| AlexNet + FC6 + LDA (*Fei et al., 2020*) | 5-fold | 95 |
| Salient facial patches based FER (*Happy & Routray, 2014*) | 10-fold | 91.80 |
| LiveEmoNet (*Podder, Bhattacharya & Majumdar, 2022b*) | 10-fold | 96.36 |
| Hybrid Methodology (*Khan et al., 2019*) | 10-fold | 92.16 |
| Dense_FaceLiveNet (*Hung, Lin & Lai, 2019*) | 5-fold | 90.97 (±3.95) |
| MobileNet with Softmax + Center loss (*Miao et al., 2019*) | 5-fold | 95.24 |
| Proposed method | 10-fold | 96.83 (±3.15) |

reveals that some sad emotions are misclassified as fear. Such misclassification is due to minor intraclass variations in subject images. Compared to different state-of-the-art methods, the proposed method gives higher accuracy, as shown in Table 3. However, it can be noticed that the standard deviation is high (±3.05) because some folds give low accuracy as the intra-class differences are very low, as shown in row 1 of Fig. 9, which makes it hard even for a human being to distinguish. Further, the dataset contains mislabeled data, as shown in row 2 of Fig. 9, which gives low accuracy in some folds. In our experiment, some folds gave 100% accuracy, but some folds gave low accuracy due to mislabeled data.

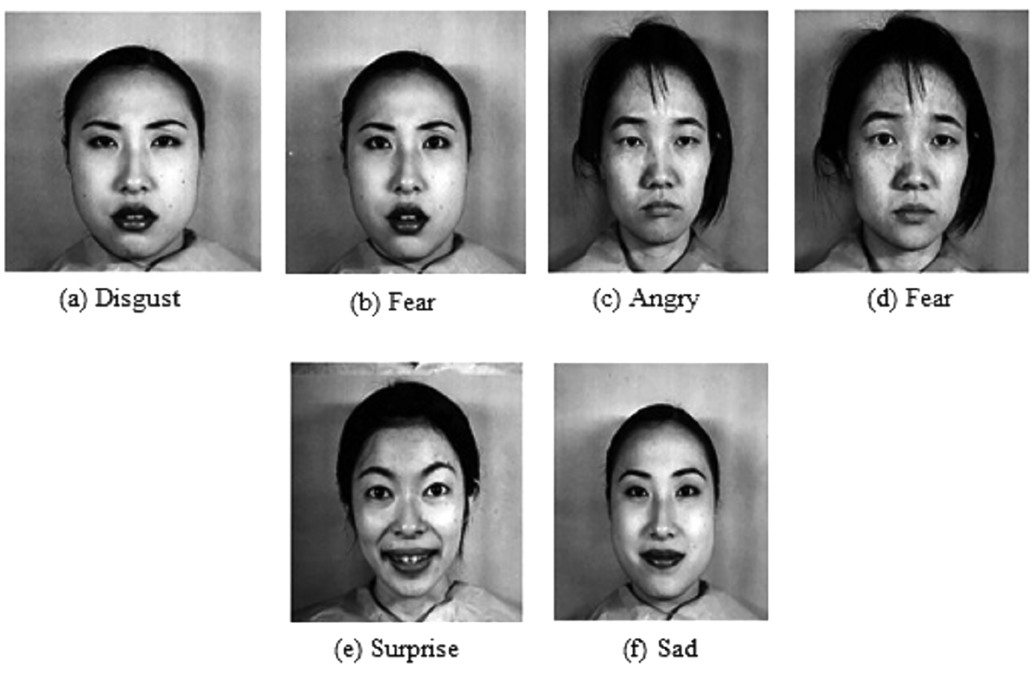

**Figure 9** Sample images from JAFFE dataset shows low intra-class variation and mislabeled images. Image source credit: the JAFFE dataset, *Lyons, Kamachi, Gyoba (2020)* and *Lyons (2021)*.

## Experimental results on CK+ dataset

The CK+ dataset contains a set of frames displaying emotional states from neutral to peak with an additional class contempt, which drives the researcher to develop different evaluation methodologies with varying classes. For the experiment, the last peak frame for six emotions and the first frame for neutral are used. Further, the five-fold cross-validation technique neutralizes errors due to random or self-selected train-test splits. The proposed method outperforms most state-of-the-art methods in terms of accuracy, as shown in Table 4. Figure 8B displays the normalized confusion matrix obtained for one fold.

## Impact of parameter, augmentation and transfer learning

The proposed method requires less parameter calculation and Giga Floating Point Operations per Second (GFLOPs) than the literature, as shown in Table 5, which is ideal for real-time deployment. Further, training the model with augmented (WA) images reduces the loss compared to training without augmented (WOA) images, as shown in Fig. 10. It is challenging for CNN to learn enough deep features from a limited dataset with small interclass variations. However, fine-tuning the proposed method with FER-2013 improves the effectiveness of the JAFFE and CK+ datasets, as can be noticed in Table 6. However, the accuracy obtained in FER-2013 is lesser than CK+ and JAFFE because the images are captured in an unconstrained environment.

**Table 4 Performance comparison with literature on the CK+ dataset.**

| Methods | Evaluation setup | Accuracy (%) |
|---|---|---|
| AlexNet + FC6 + LDA (*Fei et al., 2020*) | 7 classes | 94.70 |
| Active learning + VGG16 (*Ahmed et al., 2018*) | 7 classes | 91.80 |
| Pre-processing + CNN (*Lopes et al., 2017*) | 7 Classes | 95.75 |
| Lightweight CNN (*Podder, Bhattacharya & Majumdar, 2022a*) | 7 Classes | 96.12 |
| Light-CNN (*Shao & Qian, 2019*) | 7 Classes | 92.68 |
| Dual-branch CNN (*Shao & Qian, 2019*) | 7 Classes | 85.71 |
| Pre-trained CNN (*Shao & Qian, 2019*) | 7 Classes | 95.29 |
| LiveEmoNet (*Podder, Bhattacharya & Majumdar, 2022b*) | 7 Classes | 96.55 |
| MobileNet with Softmax + Center loss (*Miao et al., 2019*) | 7 Classes | 95.38 |
| Proposed method | 7 Classes | 96.73 (±1.45) |

**Table 5 Comparison of state-of-the-art CNN architectural parameters.**

| CNN architectures | Number of parameters (in million) | Number of GFLOPs |
|---|---|---|
| AlexNet + FC6 + LDA (*Fei et al., 2020*) | 41.5 | 0.7 |
| SCAN (*Gera & Balasubramanian, 2021*) | 70 | 7 |
| MobileNet with Softmax + Center loss (*Miao et al., 2019*) | 3.2 | 0.4 |
| Dense_FaceLiveNet (*Hung, Lin & Lai, 2019*) | 15.3 | – |
| DenseNet201 (*Huang et al., 2017*) | 5.5 | 8.5 |
| VGG16 (*Simonyan & Zisserman, 2014*) | 138 | 15.3 |
| InceptionV3 (*Szegedy et al., 2016*) | 24 | 6 |
| Proposed CNN model | 2.8 | 0.3 |

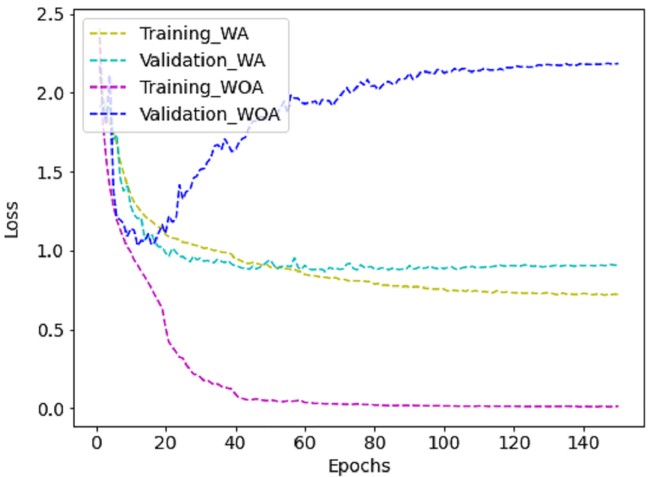

**Figure 10 The training *vs* validation loss graph on FER-2013 dataset with augmentation (WA) and without augmentation (WOA).**

**Table 6 Accuracy comparison through transfer learning from FER-2013.**

| Datasets | Without fine-tuning | With fine-tuning |
|---|---|---|
| CK+ | 89.41 (±1.68) | 96.16 (±3.05) |
| JAFFE | 73.64 (±1.87) | 96.98 (±1.45) |

**Table 7 Comparison of computational time for FER in real-time.**

| Process followed | Specifications of the system | Emotion classification time (ms/frame) | Pre-processing time (ms/frame) | Face detection time (ms/frame) | Total time (ms) |
|---|---|---|---|---|---|
| Pre-processing + CNN (Lopes et al., 2017) | NVIDA GeForce GTX 660 GPU | 10 | 20 | – | – |
| MobileNet with Softmax + Center loss (Miao et al., 2019) | NVIDIA Quadro K4200 GPU | 3.57 | 7.49 | 46.93 | 57.99 |
| Proposed method | intel i7 2.60 GHz CPU | 9.27 | 1.02 | 29.12 | 39.41 |

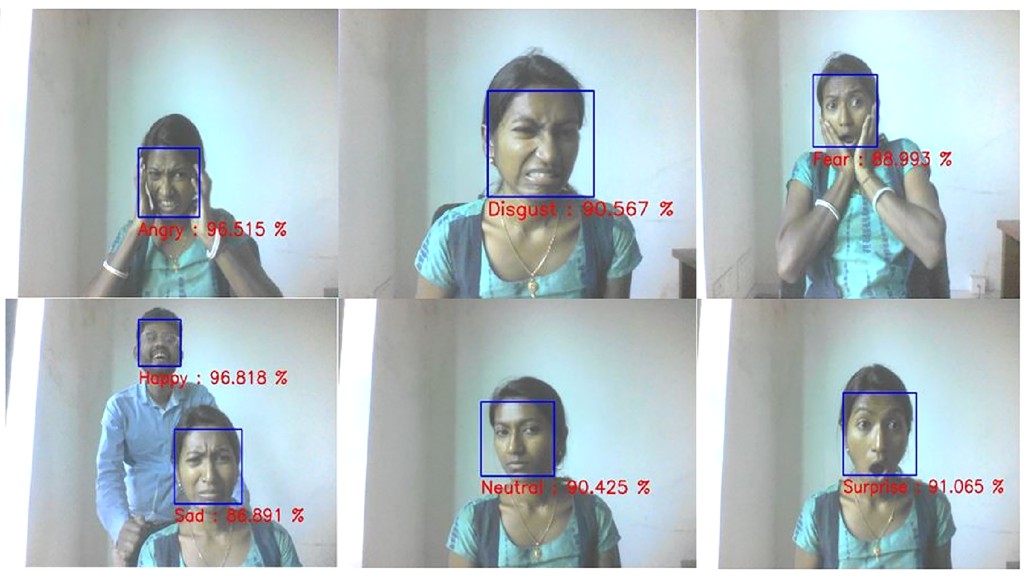

**Figure 11 Examples of Real time FER.**

## Experimental results on real-time application

We developed a webcam-based application on an Intel i7 3.47 GHz PC without GPU support to evaluate the proposed system's performance in detecting real-time facial expressions. The greyscale conversion of a frame and face detection takes approximately 29.12 ms/frame by utilizing different Python packages (*e.g.*, cv2, Keras, time, *etc.*). Further, the pre-processing (resizing, normalization, *etc.*) and prediction module take 1.02 and 9.27 ms/frame, respectively. Finally, the application calculates the runtime cost by

considering an average processing time of 100 frames. Therefore, the total time for FER is approximately 39.41 ms/frame.

The proposed method provides good accuracy in classifying facial expressions with less computational time than the conventional CNN-based method (*Miao et al., 2019*; *Lopes et al., 2017*), as listed in Table 7. Figure 11 depicts the real-time prediction results of the proposed method based on a live webcam feed. The results reveal that the FER system correctly recognizes all the universal expressions with a high confidence rate.

## CONCLUSION

An approach to FER based on deep learning is proposed in this article. The performance of the proposed system is tested on both wild and lab-controlled datasets. Based on the results, it is evident that the proposed approach outperforms not only the traditional methods but also the popular deep learning methods. The real-time detection results reveal that it can detect all expressions with high accuracy. Also, it can easily be applied to resource-constrained environments since it requires fewer parameter calculations. Future improvements of the system should focus on detecting expressions of extremely non-frontal and occluded faces. Again, efforts will be made to further improve the accuracy of the model.

### Funding
This work received no funding for this work.

### Competing Interests
The authors declare that they have no competing interests.

### Author Contributions
- Tanusree Podder conceived and designed the experiments, performed the experiments, analyzed the data, performed the computation work, prepared figures and/or tables, and approved the final draft.
- Diptendu Bhattacharya performed the experiments, performed the computation work, authored or reviewed drafts of the article, and approved the final draft.
- Priyanka Majumder conceived and designed the experiments, analyzed the data, prepared figures and/or tables, and approved the final draft.
- Valentina Emilia Balas conceived and designed the experiments, performed the experiments, analyzed the data, authored or reviewed drafts of the article, and approved the final draft.

### Data Availability
The data are available at Kaggle:

- Dataset name- FER-2013, https://www.kaggle.com/datasets/msambare/fer2013, *Goodfellow et al., 2013*.

- Dataset name-CK+, https://www.kaggle.com/datasets/shawon10/ckplus, *Lucey et al., 2010*.
- Dataset name-JAFFE, https://zenodo.org/record/3451524#.YvSw1nZBzIU, *Lyons et al., 1998*.

The code files are available in the Supplemental Files.

## Supplemental Information

Supplemental information for this article can be found online at http://dx.doi.org/10.7717/peerj-cs.1216#supplemental-information.

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
