# Peer review of "A feature boosted deep learning method for automatic facial expression recognition"

_PeerJ Computer Science, doi:10.7717/peerj-cs.1216_

## Round 0.1 · original submission · Major Revisions

Revise the paper according to the comments of reviewers.

Reviewer 1 has requested that you cite specific references. You may add them if you believe they are especially relevant. However, I do not expect you to include these citations, and if you do not include them, this will not influence my decision.

Reviewer 1 ·

Basic reporting

Thanks for this writeup, but it will be a good idea to consider the following
1. Proofreading your content again: check line 18, 185, for example
2. It is not too clear how line 56-60 becomes a contribution
3. You have over 30k data in FER-2013, which will help you transfer weights to CK & JAFFE, it is unclear why augmentation is required again. You may want to verify computationally the need for augmentation in your study.
4. Be consistent with the word FER2013. In some cases it is FER-2013

Experimental design

Thanks for this design, but a few things are unclear; kindly help your audience out on this.
1. Haar Cascade is used as a preprocessing tool in line 132, but the focus of your study was on the inefficiency of existing models not being able to work well in the wild. Haar is originally defined for frontal faces, and we don't expect all faces in the wild to have such constraints, so how was haar used in your work to detect all faces accurately?
2. The statement " Initially, two Conv layer extracts 32 feature maps" on line 145 is quite ambiguous to reproduce. Are the 32 features from the two layers or one layer? Kindly clarify
3. Line 166 & 167 is not clear. It is technically difficult to access the rigour of your contribution. The proposed CNN is still traditional. A deeper elaboration will be much appreciated to help readers know what exactly you are contributing to knowledge technically and computationally in your proposed architecture.

Validity of the findings

Thanks for presenting your findings, but kindly address this for me
1. Line 201 - 204 is overly repeated. Kindly consider taking it out
2. What is the rationale for the different batch sizes stated in line 204
3. Justification for the choice of 0.001 in line 210 is missing
4. Line 211 is not contributing to the study
5. It will be unfair to conclude that your model performs better than state-of-art methods in FER-2013. The ultimate goal of this study is the wild dataset, and one will expect that under no condition should any method work better than your model, which is not so in this finding. Primarily, accuracy is rated higher than time complexity in this kind of study because we are dealing with human subjects, and false positive has to be reduced drastically. Kindly reconsider Section 5.1. If your study cannot perform better than a study conducted in 2019 (3 years from now), you may want to reconsider your main objective again.
6. Reading these articles will suggest to you that quite a lot of work is done with JAFFE with close to 100% accuracy with less computational cost
https://www.hindawi.com/journals/am/2022/7374550/
https://www.hindawi.com/journals/am/2021/4981394/
https://vciba.springeropen.com/articles/10.1186/s42492-022-00109-0

Reviewer 2 ·

Basic reporting

The authors present a generally well-structured article, however there are certain areas where the work may be improved. The most recent works on the subject from other authors should also be included. There is no page restriction, thus authors are free to increase the literature review by including more published works.

Experimental design

The proposed approach, illustrated in Figure 2, has a very simple architecture aside from the addition of the fancy term "feature boosting." Although the authors claim that the proposed design performs well, the exact reason for this is still a mystery. A feature map visualisation, like GradCam, could be useful to determine the actual reason for better performance.

Validity of the findings

The authors claim that the accuracy and time are the major comarison tools of the results of the proposed model with the state-of-the-art methods. The performance of the proposed method in comparison to state-of-the-art methods in terms of computational time, however, is not reported by the authors. Not all cases of improved results can be proven by merely reporting the number of parameters. So, authors need to justify this claim with suitable evidence by providing a graph or table.

Additional comments

1. Although the training and validation accuracy typically portray a more accurate picture of the model's performance, Figure 4 demonstrates that the model is already overfitting, as shown by the increasing validation loss. By employing state-of-the-art methods to combat overfitting other than simple data augmentation, the authors may attempt to reduce overfitting.
2. The authors assert that their feature-boosting module increases the number of relevant features, yet the phrase "Relavent features" is vague; kindly give a precise definition of the term. How the model decides which features are relavent?
3. The grammer and overall writting of the paper can also be improved, for instance, in the line number 18 (abstract) authors write " no of relavant features", it should have been " number of relevant features" instead.

---

## Round 0.2 · accepted · Accept

After modification, this paper is suitable for publication in this journal.

Reviewer 1 ·

Basic reporting

The updated article is revised accordingly

Experimental design

The updated article is revised accordingly

Validity of the findings

The updated article is revised accordingly

Additional comments

N/A

Reviewer 2 ·

Basic reporting

The authors present a deep learning-based FER approach with minimal parameters, they claim to give better results for lab-controlled and wild datasets. Their method uses features boosting module with skip connections which help to focus on expression-specific features. The paper has sufficiently structured to be published.

Experimental design

No Comments

Validity of the findings

The results are now structured and presented in a better way.

Additional comments

No comments